

# Characterizing natural variability in complex hydrological systems using Passive Microwave based Climate Data Records: a case study for the Okavango Delta

*Robin van der Schalie[1*], Mendy van der Vliet[1], Clément Albergel[2], Wouter Dorigo[3], Piotr Wolski[4] and Richard de Jeu[1]*

[1]VanderSat B.V., Water and Climate Unit, Haarlem, Netherlands

[2]European Space Agency Climate Office, ECSAT, Harwell Campus, Didcot, Oxfordshire, UK

[3]CLIMERS, TU Wien, Department of Geodesy and Geoinformation, Vienna, Austria

[4]Climate System Analysis Group, University of Cape Town, Cape Town, South Africa

**\* Correspondence:** Robin van der Schalie, rvanderschalie@vandersat.com

**Keywords: Okavango Delta, Passive Microwave Observations, Climate Data Records, Interannual Variability, Surface Soil Moisture, Land Surface Temperature, Vegetation Optical Depth, Land Parameter Retrieval Model**

## Abstract

The Okavango river system in southern Africa is known for its strong interannual variability of hydrological conditions. Here we present how this is exposed in surface soil moisture, land surface temperature, and vegetation optical depth as derived from the Land Parameter Retrieval Model using an inter-calibrated, long term, multi-sensor passive microwave satellite data record (1998-2020). We also investigate how these interannual variations relate to state-of-the-art climate reanalysis data from ERA5-Land. We analyzed both the upstream river catchment and the Okavango Delta, supported by independent data records of discharge measurements, precipitation and vegetation dynamics observed by optical satellites. The seasonal vegetation optical depth anomalies have a strong correspondence with MODIS Leaf Area Index (correlation catchment: 0.74, Delta: 0.88). Land surface temperature anomalies derived from passive microwave observations match best with those of ERA5-Land (catchment: 0.88, Delta: 0.81), as compared to MODIS nighttime LST (catchment: 0.70, Delta: 0.65). Although surface soil moisture anomalies from passive microwave observations



and ERA5-Land correlate reasonably well (catchment: 0.72, Delta: 0.69), an in-depth

evaluation over the Delta uncovered situations where passive microwave satellites record strong fluctuations, while ERA5-Land does not. This is further analyzed using information on inundated area, river discharge and precipitation. The passive microwave soil moisture signal demonstrates a response to both the inundated area and precipitation. ERA5-Land however, which by default does not account for any lateral influx from rivers, only shows a response to

the precipitation information that is used as forcing. This also causes the reanalysis model to miss record low land surface temperature values as it underestimates the latent heat flux in certain years. These findings demonstrate the complexity of this hydrological system and suggest that future land surface model generations should also include lateral land surface exchange. Also, our study highlights the importance of maintaining and improving climate

data records of soil moisture, vegetation and land surface temperature from passive microwave observations and other observation systems.

# 1    Introduction

Long-term data records of key components of the climate system, known as essential climate

variables (ECV), are important for improving our understanding and predictability of climate behavior at different time scales (Hollmann et al., 2013; Bojinski et al., 2014). These records can help us to determine the root causes of observed climate change, e.g. natural or anthropogenic, assess its impacts and associated risks, and support mitigation and adaptation activities. In 2008, the European Space Agency (ESA) started the Climate Change Initiative

programme (CCI) to develop these ECVs from satellite data records. This was done in response to the The United Nations Framework Convention on Climate Change (UNFCCC) need for systematic monitoring of the climate system. Today, the CCI programme covers 21 satellite-based ECV records  (Projects (esa.int), last visited September 2021).

Surface soil moisture (SSM) is one of these ESA CCI ECVs. These records are based on a fusion of both passive and active microwave satellite retrievals (Dorigo et al., 2017). The current version 6.1 spans from 1979 until 2020 (Scanlon et al., 2021), and contains three separate SSM products, which are derived from active, passive, and a combination of active and passive sensors. The methodology and evaluation of the harmonisation and merging of



the soil moisture retrievals from multiple satellites is described by Gruber et al. (2019). ESA CCI SSM data has been used for more than 10 years as the baseline for the annual evaluation and interpretation of global SSM conditions as reported in the leading BAMS' "State of the Climate" reports (Van der Schalie et al., 2021). Three datasets are produced as part of the passive input for the ESA CCI SM, which is SSM ($SSM_{MW}$), but also land surface temperature ($LST_{MW}$), and vegetation optical depth ($VOD_{MW}$).

$SSM_{MW}$ data sets have been extensively evaluated with ground observations, models, other satellite products, and related ECVs like precipitation (e.g. Hirschi et al., 2021; Beck et al., 2021; Dorigo et al., 2015; Al-Yaari et al., 2019; Albergel et al., 2013; Loew et al., 2013). $VOD_{MW}$ has been used in multiple studies with a focus on seasonal and interannual vegetation dynamics (e.g. Liu et al., 2015; Moesinger et al., 2020; Teubner et al., 2019) or specifically on L-band VOD characteristics (e.g. Schwank et al., 2021; Bousquet et al., 2021; Rodriguez-Fernandez et al., 2018). Research on the quality of $LST_{MW}$ (e.g. Holmes et al., 2009; Holmes et al., 2015) remains limited. The robustness of the interannual variability signals within these multi-decadal data records is still not always clear, and a combined assessment of all three variables is necessary for understanding these datasets, as the current joint retrieval algorithm make their values fundamentally intertwined (Owe et al., 2008). Such information provides unique opportunities for both monitoring and seasonal forecasting, e.g. over Africa (e.g. Cook et al., 2021).

The purpose of this paper is to improve insight into the interannual signals of the $SSM_{MW}$, $LST_{MW}$ and $VOD_{MW}$ by presenting a case study over a region with a complex hydrological system, i.e. the Okavango, and how their skill compares to state-of-the-art climate reanalysis data from ERA5-Land (Muñoz-Sabater, 2019; Muñoz-Sabater, 2021). ERA5-Land aims to quantify the water and energy cycles over land in a consistent manner, therefore allowing the characterisation of trends and anomalies. Although ERA5-Land (E5) data is known to be of high quality in many regions around the globe, for use in any specific regions this needs to be properly evaluated. Therefore, this dataset does not only function as a benchmark in this study, but will also be analyzed in more detail to evaluate its ability to properly detect the natural dynamics and variability in the Okavango and how this compares to the signal of the passive microwave-based datasets. Other datasets are used as support for determining which dataset (i.e., either PMW or E5L estimates of the same variable) is more likely to reflect true



conditions. This research can help to improve the synergy between EO data sets and land surface models, and to identify both strengths and shortcomings of either one.


More specifically, the Okavango Delta and Okavango River Catchment in southern Africa were selected as the study area. The Okavango Delta (Republic of Botswana, 2013) consists of permanent marshlands and seasonally flooded plains, and is one of the few endorheic "delta" systems (geomorphologically Okavango Delta is an alluvial fan, Kgathi et al. 2006)

that does not flow into the ocean. It is an exceptional example of the interaction between climatic, hydrological and biological processes, leading to a unique mix of flora and fauna, and has therefore been included in the UNESCO World Heritage List since 2014. Three features in the local hydrological system stand out, i.e., the strong interannual variability, the lateral water influx component of the Okavango River into the Delta, and the seasonal

characteristics with a lag between rainfall, river discharge and flooding. Unfortunately, it is expected that global warming will affect this natural variability in the hydrological cycle over the Okavango Delta (Wolski et al., 2014; Wolski et al., 2012), for example reducing high-water periods like in 2009-2011. These kinds of negative impacts increase the need for reliable monitoring capabilities.


The structure of the paper is as follows. Chapter 2 introduces the study area and includes the exact regions of interest (ROIs) that are used for the data extraction. Sect. 3.1 describes the passive microwave data and other data sources. Sect. 3.2 explains the methodology, concerning the inter-calibration (3.2.1), LPRM (3.2.2), evaluation of the dataset anomalies

(3.2.3.), and of the river, flood and precipitation contribution to SSM anomalies over the Okavango Delta (3.2.4). Chapter 4, 5 and 6 provide the Results, Discussion and Conclusions of these different steps.

## 2   Research Area

With a length of approximately 1600 km, the Okavango river is one of the largest in southern Africa (Muzungaire et al. 2012). The river is known globally for its large terminating inland "delta". The Okavango Delta is a large seasonally pulsed inland wetland, a mixture  of aquatic vegetation, open water, and dry land with the actively inundated area covering a part of the 28,000 km$^2$ alluvial cone (Ringrose et al., 1988).






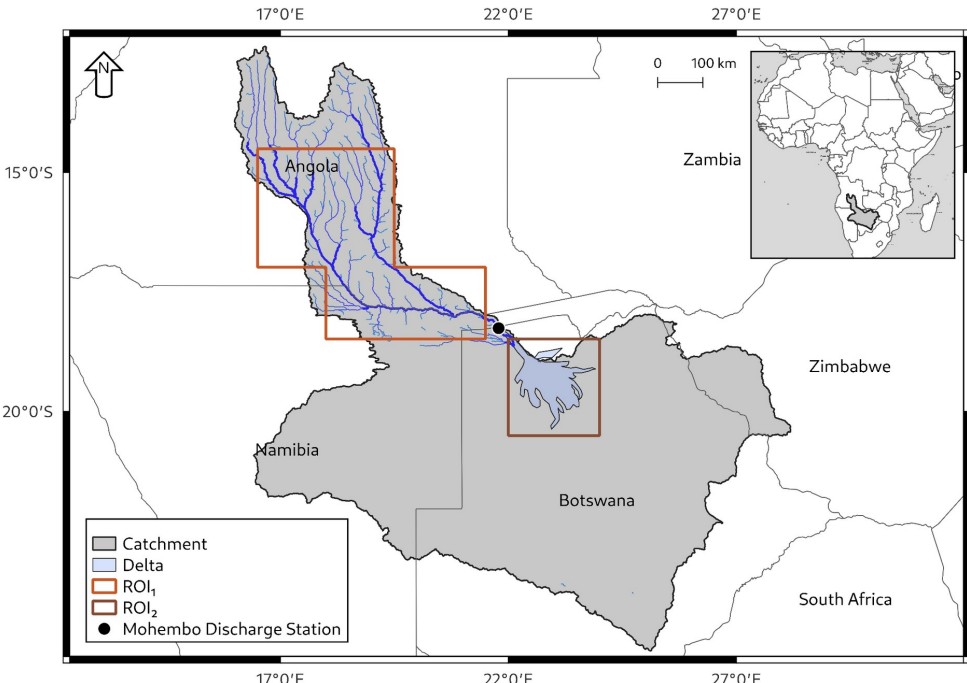

**Figure 1: The research area comprising ROI$_1$ (a part of the upstream area of the Cubango and the Cuito River) and ROI$_2$ (the surrounding of the Okavango Delta) in relation to the Okavango drainage basin (grey). The black dot marks the location of the discharge station at Mohembo.**


In line with both the interannual variation in local and upstream rainfall and the longer-term effects of surface-groundwater interactions, substantial interannual variability in the Delta's inundated area was recorded over the period 1932-2000 (Wolski and Murray-Hudson, 2008), with annual minima of about 3000 km$^2$ up to annual maxima of 12000 km$^2$ (Wolski et al.,

2017; Gumbricht et al., 2004). Whereas estimates for the total annual water budget stemming from direct rainfall in the Okavango Delta ranges between 25% to 50%, the Okavango River inflow accounts for the other 50% to 75% (McCarthy et al., 1998; McCarthy et al., 2000; Ashton and Manley, 1999; Ashton and Neal, 2003, Wolski et al., 2006).

In this study we focus on only two perennial rivers in the Okavango catchment - the Cubango River and the Cuito River (Ashton and Neal, 2003). Data was extracted from the catchment area within ROI$_1$ of Figure 1. These rivers originate in Angola and are a vital lifeline to the

Okavango Delta with an average inflow at Mohembo of 9863 Mm3 in the period 1932-2001
and a 71.4% contribution to the total water budget of the Delta.


The Angolan part of the basin is characterized by a subtropical climate, while in Botswana
and Namibia parts are classified as semi-arid (Kgathi et al. 2006). During drought years in the
1980s and 1990s, the annual inflow at Mohembo reduced up to 45% (McCarthy et al., 2000;
Ashworth, 2002; Ashton, 2003; Ashton and Neal 2003) which then coincided with

proportional declines of the Okavango Delta outflow to the Thamalakane and Boteti rivers
(Ashton & Manley 1999; Ashworth 2002, Ashton and Neal 2003). Throughout these periods
a growing demand for water arose in Botswana and Namibia (MGDP, 1997; Ashton, 2003).
Overall, the dry phase was caused by multi-decadal oscillations in rainfall, and likely related
to processes of internal variability in the climate system (Wolski et al., 2012).


$ROI_1$ and $ROI_2$ were chosen to study how their significantly different water influxes affect the
signal of the data sources used in the evaluation. The Delta is of particular interest, as it is
mostly driven by a strong and highly variable lateral influx from the Okavango River that
creates a pattern of seasonally varying wetness that is asynchronous or off-phase with the

rainy season.

## 3   Material and Methods

### 3.1  Data

#### 3.1.1  Passive microwave observations

The three main variables that are used for the analysis are surface soil moisture ($SSM_{MW}$),
vegetation optical depth ($VOD_{MW}$) and land surface temperature ($LST_{MW}$). These variables are
derived from passive microwave observations from multiple satellite sensors that observe in
similar frequencies and overlap in time.

The Advanced Microwave Scanning Radiometer for EOS (AMSR-E, Kawanishi et al., 2003)
is a twelve-channel, six-frequency, passive microwave radiometer developed by the Japan
Aerospace Exploration Agency (JAXA) and was active between 2002 and 2011. AMSR-E is



part of the payload carried onboard the Aqua (EOS PM-1) NASA scientific research satellite, which has a polar orbit with a 1:30 pm / am equatorial crossing time for ascending / descending swaths. AMSR-E was launched to obtain data to improve our understanding of global-scale water and energy cycles and played a key role in the development of soil moisture retrieval algorithms. For the technical specifics, see Table 1. Only descending brightness temperature data was used for this study.

The Advanced Microwave Scanning Radiometer 2 (AMSR2, Imaoka et al., 2012) onboard the GCOM-W1 satellite is the follow-up of AMSR-E, and was launched in 2012. Although incorporating improvements, the general setup is similar to AMSR-E (see Table 2). However, unfortunately there is a gap between AMSR-E and AMSR2 of about 9 months, making a direct intercalibration of time series complicated.

To overcome this gap and to extend the passive microwave observation record back to 1998, we make use of the Tropical Rainfall Measuring Mission's (TRMM, Kummerow et al., 1998) Microwave Imager (TMI). TMI observes in X-band and higher frequencies. TRMM is not in a polar orbit because of its focus on the Tropical regions and therefore does not cover the entire globe. Data is only available between 40°N and 40°S and due to its orbital characteristics has a variable crossing time, see Table 1. Only brightness temperature data was used that had a local overpass time between 10:30 pm and 4:30 am, to best match AMSR-E and AMSR2.

For this study we use X-band brightness temperature data due to its availability on all three sensors. Ka-band is the main frequency used for the $LST_{MW}$. All brightness temperatures were collected and gridded into a 0.25° grid for the Okavango Delta area.





**Table 1: Overview and characteristics of passive microwave satellite sensors used in the study.**

| Sensor | Provider | Temporal coverage | Bands | Spatial coverage | Swath Width | Equatorial crossing time | Data level |
|---|---|---|---|---|---|---|---|
| Advanced Microwave Scanning Radiometer for EOS (AMSR-E) on AQUA | JAXA / NASA | 07/2002 – 10/2011 | C, X, Ku, K, Ka, W | Global | 1445 km | Asc: 13:30 Desc: 1:30 | L2A v3 |
| Advanced Microwave Scanning Radiometer 2 (AMSR2) on GCOM-W1 | JAXA / NASA | 05/2012 – ongoing | C, X, Ku, K, Ka, W | Global | 1450 km | Asc: 13:30 Desc: 1:30 | L1R |
| Tropical Rainfall Measuring Mission's (TRMM) Microwave Imager (TMI) | NASA | 01/1998 – 12/2013 | X, Ku, K, Ka, W | N40° to S40° | 780 or 897 km after orbit boost 8/2001 | Varies (non polar-orbit) | L1C (XCAL, Berg et al., 2016) |

### 3.1.2 Ancillary data sets

In our analysis we use several ancillary data sets to determine the ability of passive microwave-based satellite data records to correctly capture interannual variations. These ancillary datasets are split into two types:

Firstly, data was used from the ERA5-Land climate reanalysis model (Muñoz-Sabater, 2019; Muñoz-Sabater, 2021), which is an enhanced resolution (9 km x 9 km) land-only offline

rerun of the ECMWF ERA5 climate reanalysis (Hersbach et al., 2020). $SSM_{E5}$, $LST_{E5}$ and $PR_{E5}$ were extracted. For both $SSM_{E5}$ and $LST_{E5}$ the Layer 1 (0-7cm) was used. $LAI_{E5}$ was excluded from the analysis as it only contained a climatology based on satellite EOs (no interannual variability). ERA5-Land data was extracted from the Copernicus Climate Change Service (C3S) Climate Data Store (CDS). As it has an hourly resolution, the values closest to

the satellite overpasses were chosen. Data covers the complete period of 1998 to 2020.

Secondly, independent observational datasets are used, which have the sole purpose of functioning as a benchmark. These consist of the Okavango River Discharge measurements (ORD, Okavango Research Institute, 2021), Okavango Delta Inundated Area ($ODIA_{MD}$), Leaf Area Index ($LAI_{MD}$, Yang et al., 2006) and nighttime LST ($LST_{MD}$, Wan, 2014) from the

Moderate Resolution Imaging Spectroradiometer (MODIS), and precipitation from the NASA Global Precipitation mission's IMERG product ($PR_{IM}$, Huffman et al., 2015).





A majority of the water entering the Okavango Delta originates from the Delta inlet at Mohembo. Therefore, we use ORD from the Mohembo station (see Fig. 1) to indicate the long term variability of the lateral inflow into the Delta. Measurements, using E-type gauge

plates, are done on a regular (fortnightly) basis by the Botswana Department of Water Affairs, and the data are shared by the Okavango Research Institute of the University of Botswana. The advantage of using this data set is that it has a long historical record dating back to 1974. For this study, data was extracted for the 1998 to 2020 period.

$ODIA_{MD}$ represents the inundated area in the Okavango Delta, and is derived from using

shortwave infrared (SWIR) observations from the MODIS sensor (Wolski et al., 2017). More specifically data for band b7 from the MCD43A4 product was used. Reflectances of training areas are used to dynamically determine the threshold used for the derivations of the inundation. An automated and up to date monitoring tool for the flooding extent can also be found online (http://www.okavangodata.ub.bw/).

The $LAI_{MD}$ is defined as the one-sided green leaf area per unit ground area (Chen et al., 1992; Yang et al., 2006). The $LAI_{MD}$ for the study area, including both the drainage Catchment and the Delta, was extracted from the MOD15A2H Version 6 MODIS dataset. This is an 8-daily product that uses the best available pixel within the 8-day period. The product has a spatial resolution of 500 m, and the mean was extracted for the complete ROIs.

1 km nighttime, about 1:30 am, surface temperature from MODIS was extracted from the MYD11A2.006 product, which is based on the average over 8 days of all available $LST_{MD}$ observations. For this study the mean values of the two areas were extracted. The temporal coverage is from February 2000 to the end of 2020 for the $LAI_{MD}$ and July 2002 to the end of 2020 for the $LST_{MD}$.

For $PR_{IM}$, data was used from the Integrated Multi-satellitE Retrievals for GPM (IMERG, Huffman et al., 2015), which is produced at 0.1° resolution. IMERG is a unified algorithm that provides rainfall estimates based on a combination of observations from multiple passive-microwave sensors, infrared sensors and precipitation gauges. Mean daily data was used from the GPM_3IMERGDF version 6, covering June 2000 to December 2020.




## 3.2 Methods

### 3.2.1 Intercalibration of PMW brightness temperatures

The intercalibration of AMSR-E, AMSR2 and TRMM is based on the methodology described in Van der Schalie et al. (2021). In this approach a two-step linear regression model
is used, which first defines a global slope and afterwards a local intercept. Secondly, it uses a cost function that not only minimizes the differences between brightness temperatures of the individual polarizations, i.e. vertical (V) and horizontal (H), but also for the ratio between the two. This is because the Land Parameter Retrieval Model (LPRM, see next section) used for the $SSM_{MW}$, $VOD_{MW}$ and $LST_{MW}$ retrievals is very sensitive to the polarization ratio.
Inconsistencies in this ratio between different sensors can lead to an imbalance in how the radiative transfer model distinguishes between the emission from the soil and the from vegetation, respectively, leading to biases in the resulting retrievals.

This intercalibration methodology, previously applied only to the Ku-, K- and Ka-band, is here also used for the X-band data. After retrieving $SSM_{MW}$, $VOD_{MW}$ and $LST_{MW}$ from the
intercalibrated individual sensors, a linear regression is applied between the different sensors using their respective overlap. This is done to remove any inconsistencies. The improved inter-calibration between sensors can lead to a reduced need for break corrections (e.g. Preimesberger et al., 2020) and help to better address related issues at the source.

As this study focuses on anomalies at a seasonal timescale, the temporal coverage obtained
by the current three sensors is sufficient. However, as was shown by Van der Schalie et al. (2021) and as is done for the passive microwave based data input for the ESA CCI SM, other sensors like GPM, FengYun-3B and FengYun-3D can be included without issues, resulting in improved revisit times and coverage.

### 3.2.2 Land Parameter Retrieval Model

The Land Parameter Retrieval Model (LPRM, Owe et al., 2008) is a retrieval algorithm that simultaneously solves for $SSM_{MW}$, $VOD_{MW}$ and $LST_{MW}$ without the use of any ancillary data sources on vegetation or temperature. The model is based on the tau omega (τ-ω) model (Mo et al., 1982), which simulates the top-of-the-atmosphere brightness temperatures by modeling



the individual contribution of the soil, vegetation and atmosphere. LPRM mainly distinguished itself from other algorithms through the analytical derivation of the VOD (Meesters et al., 2005) and the use of Ka-band observations for the $LST_{MW}$ (Holmes et al. 2009). Here we use the latest version of LPRM developed by Van der Schalie et al. (2017).

LPRM is currently the main algorithm used for all the passive microwave-based SSM
retrievals in ESA CCI SM (Dorigo et al., 2017). Due to its unique analytical solution for the derivation of $VOD_{MW}$ that uses no external source of information for the vegetation, LPRM has also been used in several studies of long term vegetation dynamics (Liu et al., 2012; Liu et al., 2015), land degradation (Liu et al., 2013; Van Marle et al., 2017) and the development of a climate data record of $VOD_{MW}$ (VODCA, Moesinger et al., 2020).


### 3.2.3 Evaluation of anomalies

To have a better understanding of the quality of the different datasets in detecting interannual variability and anomalies, a two-step comparison analysis is done. First, the anomalies are visualized over time and their dynamics assessed. Second, the relations between related
datasets are quantified using correlations and visualized using scatter plots. This is done separately for the catchment and the delta.

The $SSM_{MW}$ is compared to the $SSM_{E5}$, both representative for the moisture conditions in the first few centimeters of the soil. As this is a direct comparison, in this step the focus will be on their similarity and differences, without analysing what causes it. Additionally, an
extensive analysis is conducted (Section 3.2.4) to determine which of the data sets most likely reflects the ground conditions, based on their relation to ORD, $ODIA_{MD}$ and PR.

For $VOD_{MW}$ there is a comparison with another regularly used satellite-based datasat, $LAI_{MD}$. Theoretically the $VOD_{MW}$ represents the attenuation of the microwave emission through the vegetation cover, which is related to both the structure and moisture content of the vegetation.
The $LAI_{MD}$ is representative of the projected single-sided green leaf area per unit ground area. Although VOD and LAI are fundamentally different, it is assumed that for dynamic and sparsely to moderately vegetated regions, i.e. excluding forests, the X-band also mostly measures the response of the leaves with the microwave signal via the vegetation water content (Jackson & Schmugge, 1991). Further defining the quality and ability of $VOD_{MW}$ to





detect interannual variability can be especially useful in improving the applicability and understanding of independent vegetation data records based on passive microwave observations like VODCA (Moesinger et al., 2020).

Here the anomalies of LST from three different sources, e.g. passive microwave ($LST_{MW}$), model ($LST_{E5,}$) and thermal infrared ($LST_{MD}$), which all represent the skin temperature of the
land surface - are evaluated.

Because the focus is on the (seasonal) variability over a multi-decadal timespan, a 91 day moving average (±45 days) is first applied to the data sets. The climatology for the anomaly calculation is based on the 2003-2020 period, as the $LST_{MD}$ is only available from 2003 onwards and overall consistency for the baseline is preferred. As the window for the moving
average is 91 days, little impact is assumed from data loss due to cloud cover in the MODIS datasets.

It is worth keeping in mind that none of these datasets provide the "truth" or measure exactly the same quantity, therefore differences are to be expected. In the analysis component (see following Section), extra attention will be given to a specific case in the Okavango Delta
where a clear divergence is observed between the different SSM datasets.

### 3.2.4 Analysis of river flooding and precipitation contribution to soil moisture anomalies in the Okavango Delta

As further in this study (Sect 4.1) the signal of the two SSM data sources ($SSM_{MW}$ and
$SSM_{E5}$) is shown to diverge over the Okavango Delta, an in-depth analysis is set up to explain the main drivers of their respective signals. This can help to better understand what the SSM data sets represent and give users insight in how to use them for their research activities and applications.

A first step in this is to directly compare the SSM data sets to the ORD, $ODIA_{MD}$ and both
$PR_{E5}$ as $PR_{IM}$. These data sets can provide insight into what is the driver of the SSM anomalies in this region. As described in section 2, about 50%-75% of the total influx of water into the Okavango Delta comes from the ORD, while PR on average contributes 25%-50%, so we expect to see this reflected in the SSM either via the ORD or the $ODIA_{MD}$ signals.





Following this, a multiple linear regression exercise is conducted. This is done to look into
the influence of the $ODIA_{MD}$, ORD and PR signals on the $SSM_{MW}$ and $SSM_{E5}$ anomalies in
the Delta. This allows us to determine the drivers of the SSM anomalies, and more
importantly, how they differ between the two. Instead of using the absolute anomalies in this
analysis, the Z-score is preferred, as this normalization removes issues with conversion of
units. A visualisation will also be made of the climatologies from the different datasets,
including their 10 and 90 percentiles, to further define the connection and time lag between
the signals of the different parameters.

# 4 Results

## 4.1 Soil Moisture

Figure 2 shows the anomalies of $SSM_{MW}$ and $SSM_{E5}$ over the Okavango catchment and Delta,
with Figure 2E/F comparing them directly to each other in a scatter plot. In both areas the two
datasets correlate moderately well, 0.717 and 0.694 respectively. In the Delta however, a
mismatch occurs in some occasions, leading to a visible flat line in the scatterplot where the
anomalies of $SSM_{MW}$ vary while the anomalies of $SSM_{E5}$ are close to 0 (Fig. 2E). The signal
$SSM_{MW}$ anomalies over the Catchment, and $SSM_{E5}$ anomalies over both the catchment and
Delta, seem to have clear short-term variability as can be seen from the peaks in the wet
season, while the dry season remains mostly stable around 0. Only the anomalies of $SSM_{MW}$
over the Delta diverge from this and show a more multi-year variation, with highs in the
years around 2011 and lows in the early and late periods of the time period. These cases will
be further analyzed in Sect. 4.4 in combination with the ORD and PR.

The absolute range of the anomalies differs to some extent between the two products: $SSM_{MW}$
anomalies range between -0.03 and 0.025 $m^3m^{-3}$ in the Catchment and -0.05 and 0.05 $m^3m^{-3}$
in the Delta, whereas $SSM_{E5}$ anomalies range between -0.10 and 0.06 $m^3m^{-3}$ in the Catchment
and -0.08 and 0.10  $m^3m^{-3}$ in the Delta. However, the dynamics of the signal are very similar.


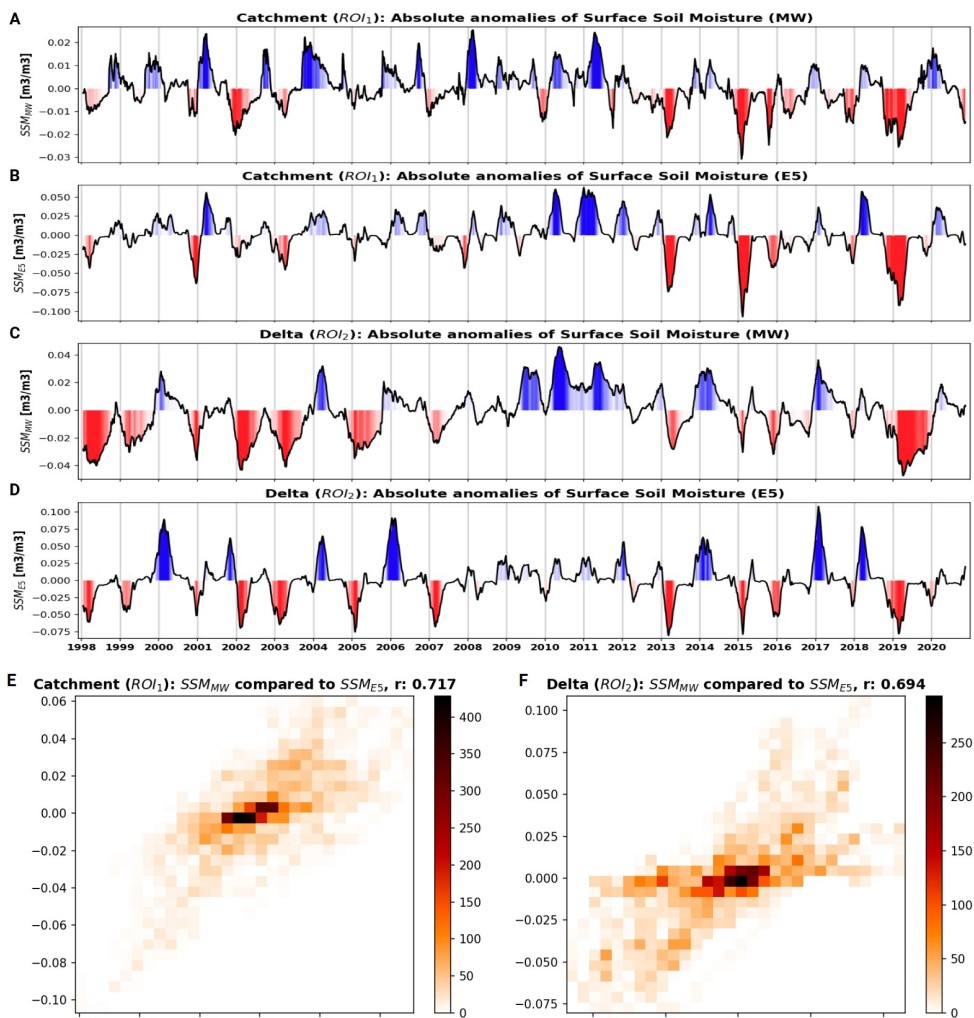

**Figure 2: SSM$_{MW}$ and SSM$_{E5}$ absolute anomalies over the Okavango Catchment (A,B,E), with the intensity of the coloring based on the z-score of the positive (blue) and negative (red) anomalies, and the Okavango Delta (C,D,F) in time series and density plots. A daily time step is used from the moving average data set.**

## 4.2 Vegetation Optical Depth

Figure 3 shows the anomalies of VOD$_{MW}$ and LAI$_{MD}$ over the Okavango Catchment and Delta, with Figure 3E/F again showing a direct comparison in a scatter plot. The two datasets have a 0.741 correlation over the Catchment and up to 0.876 in the Delta. Generally, a similar



pattern is visible for both regions. One exception can be seen during the 2008 to 2011 period
in the Catchment, where the $VOD_{MW}$ anomaly remains high throughout multiple years, while
the overall above average $LAI_{MD}$ anomalies fluctuate to a greater extent. The lowest values in
the Delta were detected early in the study period, with $VOD_{MW}$ recording an almost -0.08
anomaly during 1998 and 2003. This 2003 event is also seen in the $LAI_{MD}$ dataset, while no

data is available for 1998. In more recent years, no negative anomalies of that strength have
been recorded.

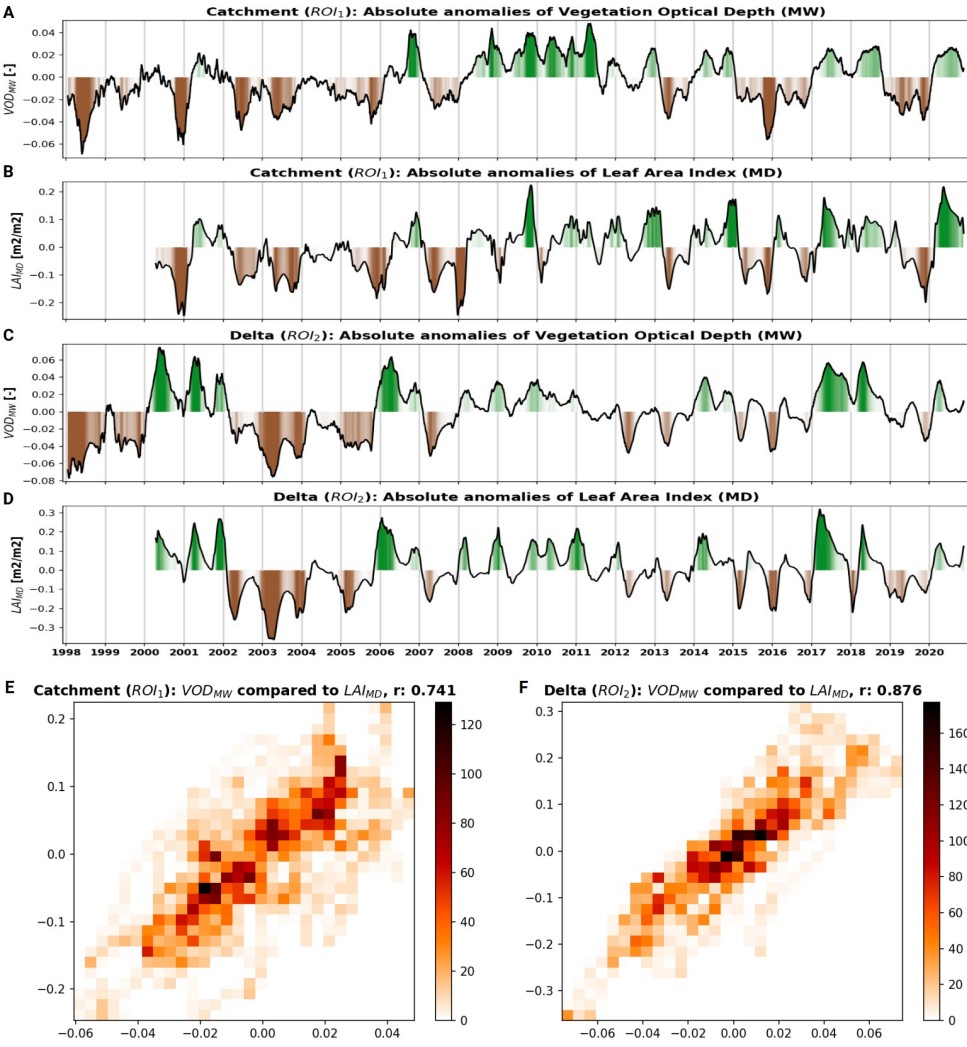



**Figure 3: VOD$_{MW}$ and LAI$_{MD}$ anomalies over the Okavango Catchment (A,B,E), with the intensity of the coloring based on the z-score of the positive (green) and negative (brown) anomalies, and the Okavango Delta (C,D,F) in time series and density plots. A daily time step is used from the moving average data set.**


## 4.3 Land Surface Temperature


Figure 4A/B shows the anomalies of LST$_{MW}$ over the Okavango Catchment and Delta, with Figure 4B/C/D/E/F/G showing a direct comparison in a scatter plot between LST$_{MW}$, LST$_{E5}$ and LST$_{MD}$. Because of the high correlation between LST$_{MW}$ and LST$_{E5}$, of 0.884 in the Catchment and 0.809 in the Delta, the decision was made to only show the LST$_{MW}$ time series to focus more on the scatterplots of the three different products. The correlation of LST$_{MW}$ against LST$_{MD}$ is much lower, with 0.643 and 0.255 for both regions, showing a low relation in the Catchment. LST$_{E5}$ compares better to LST$_{MD}$ with a correlation of 0.702 in the Catchment and 0.650 in the Delta, however this is still significantly lower than the comparison with LST$_{MW}$. The absolute ranges in the anomalies as detected by the three products are very similar.



The slightly lower correlation of LST$_{MW}$ against LST$_{E5}$ in the Delta is mostly caused by the period 2010 and 2011, when the LST$_{E5}$ anomaly (between -1 and 1 °C) is smaller than that of LST$_{MW}$ (between -3 and -1 °C). Below-average temperatures are recorded for a prolonged period between 2006 and 2014 in both regions. For the Delta, the highest temperature anomalies are recorded in 2019 and 1998. In the Catchment, this is seen in 2015 and 2019.




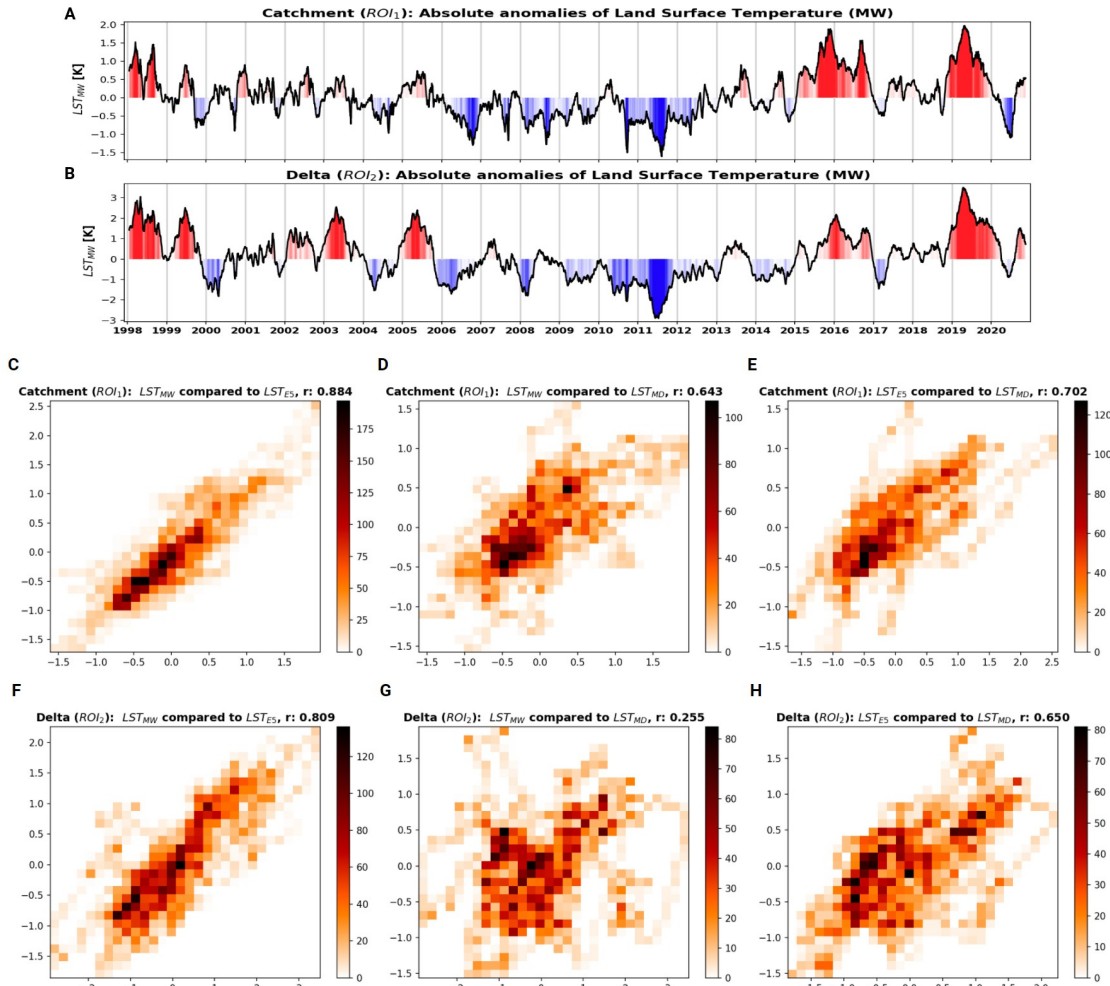

**Figure 4: LST$_{MW}$ time series over the Okavango Catchment and the Okavango Delta (A,B), with the intensity of the coloring based on the z-score of the positive (red) and negative (blue) anomalies. For the density plots; LST$_{MW}$ compared to LST$_{E5}$ (C,F), LST$_{MW}$ compared to LST$_{MD}$ (D,G), LST$_{E5}$ compared to LST$_{MD}$ (E,H). A daily time step is used from the moving average data set for the density plots.**

### 4.4 River and precipitation contribution to soil moisture anomalies in the Okavango Delta

Figure 5A, 5B and 5C show the anomalies of the ORD, ODIA$_{MD}$ and PR$_{E5}$ over the Delta, which have visibly different signals. The ORD shows a strong multi-year signal with especially high values recorded from 2009 to 2012. Outside of that period, with the exception





of 2004, values generally lay below the 2003 to 2020 climatology. The OIAD$_{MD}$ shows a signal that is relatively similar to that of the ORD, however smoother, with less variability and lagging behind. The PR$_{E5}$ over the Delta shows mostly values around 0 mm during this 2009 to 2012 period, and otherwise varies more dynamically from year to year with values

above and below the climatology.

Although SSM$_{MW}$ and SSM$_{E5}$ anomalies have an overall correlation of 0.694 in the Delta, Figure 2F shows many occasions where the SSM$_{MW}$ had negative or positive anomalies, while the SSM$_{E5}$ did not diverge from the climatology. To better assess what causes this opposite signal, the climatology (using ±15 days moving average) of different parameters are

provided in Figure 6, including their 10% and 90% percentiles. Here one can see the difference in the dynamics between SSM$_{MW}$ (Fig. 6A) and SSM$_{E5}$ (Fig. 6B). The SSM$_{E5}$ shows a clear relation to the PR datasets (Fig. 6G/H), while the SSM$_{MW}$ still picks up a moisture signal between April and September. When looking at the ORD and OIAD$_{MD}$, these are the moisture-related signals that still show strong variability in this time of the year, indicating

that the SSM$_{MW}$ could also contain information from other sources than PR. On a side note, Figure 6 shows that besides matching well with long term anomalies, LST$_{MW}$ and VOD$_{MW}$ also have a strong matching intraseasonal signal with LST$_{E5}$ and LAI$_{MD}$, respectively.

Table 2 presents the results of a multiple linear regression to determine the drivers of the observed/modelled SSM anomaly signal in the Delta, using ODIA$_{MD}$, ORD and PR as inputs.

The Z-score anomalies are used to improve the comparability between the different datasets and their weight. The results show that the weighting for SSM$_{MW}$ consists of a balance between the PR in the Delta and the ODIA$_{MD}$, with an overall slightly higher weight for the ODIA$_{MD}$, and leading to a maximum correlation of 0.843 when using PR$_{E5}$ over PR$_{IM}$. This leads to a RMSE of about 0.44 for the Z-score. The SSM$_{E5}$ anomalies are clearly, driven by

the PR$_{E5}$ anomalies, reaching a correlation of 0.866. The correlation strongly decreases to 0.64 when the PR$_{E5}$ is replaced with PR$_{IM}$, which reflects back in the RMSE of the Z-score anomalies, which increases from about 0.37 to 0.57.

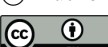

**Table 2: Results of the multiple linear regression for estimating the relationships between the Z-score anomalies of SSM, PR, ORD, and ODIA$_{MD}$.**

| Prediction | Input 1 | Input 2 | Correlation | RMSE | Weight input 1 | Weight input 2 |
|---|---|---|---|---|---|---|
| SSM$_{MW}$ | PR$_{E5}$ | ORD | 0.780 | 0.490 | 0.527 | 0.575 |
| | PR$_{IM}$ | ORD | 0.704 | 0.499 | 0.485 | 0.517 |
| | PR$_{E5}$ | ODIA$_{MD}$ | 0.843 | 0.431 | 0.436 | 0.670 |
| | PR$_{IM}$ | ODIA$_{MD}$ | 0.806 | 0.437 | 0.396 | 0.668 |
| SSM$_{E5}$ | PR$_{E5}$ | ORD | 0.866 | 0.370 | 0.878 | 0.166 |
| | PR$_{IM}$ | ORD | 0.636 | 0.571 | 0.735 | 0.163 |
| | PR$_{E5}$ | ODIA$_{MD}$ | 0.866 | 0.376 | 0.880 | 0.162 |
| | PR$_{IM}$ | ODIA$_{MD}$ | 0.646 | 0.564 | 0.714 | 0.190 |

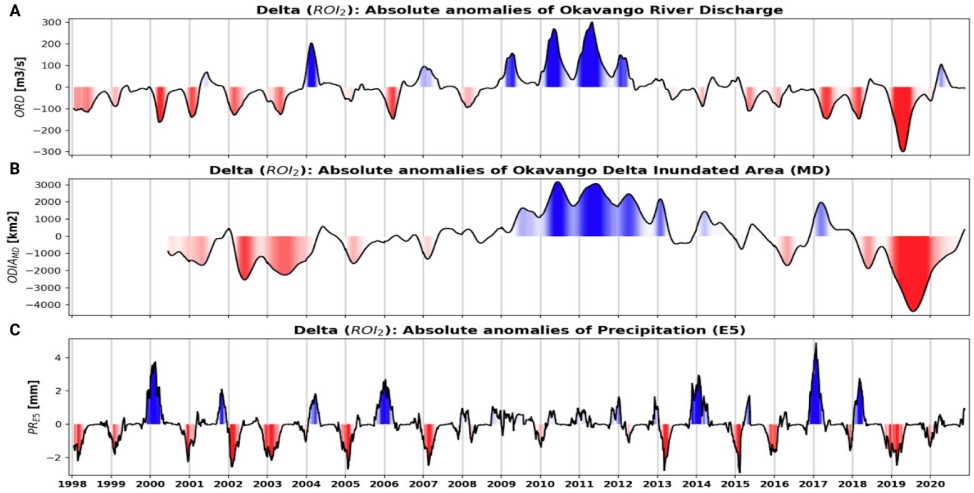

**Figure 5: ORD, ODIA$_{MD}$ and PR$_{E5}$ time series over the Okavango Delta (A,B,C), with the intensity of the coloring based on the z-score of the positive (blue) and negative (red) anomalies.**



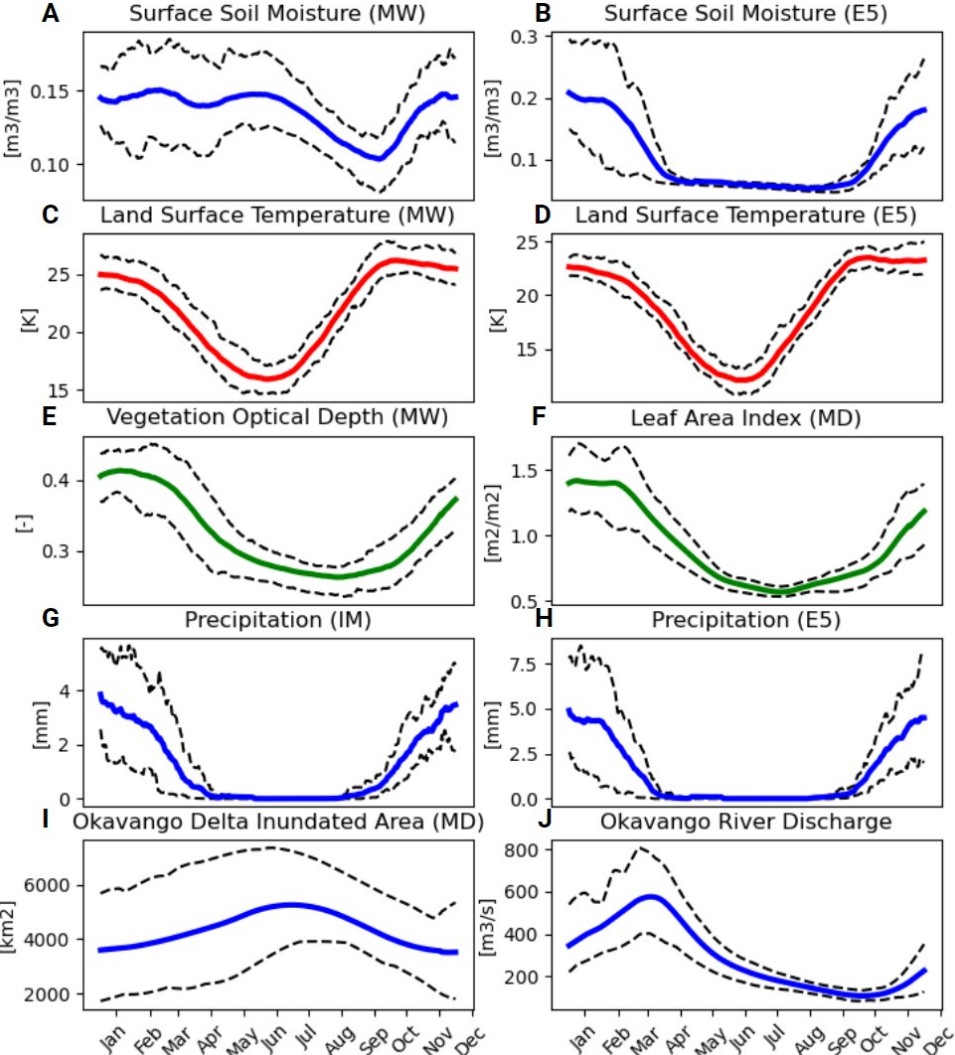

**Figure 6: Mean climatology (colored thick line) and both 10% and 90% percentiles (black dashed lines)**
**for SSM$_{MW}$ (A), SSM$_{E5}$ (B), LST$_{MW}$ (C), LST$_{E5}$ (D), VOD$_{MW}$ (E), LAI$_{MD}$ (F), PR$_{IM}$ (G), PR$_{E5}$ (H), ODIA$_{MD}$ (I)**
**and ORD (J). Data using a ±15 days moving average was plotted to distinguish between intraseasonal**
**signals.**






## 5    Discussion

Over both the Delta and the Catchment,  a remarkably strong relationship between the $LAI_{MD}$ and $VOD_{MW}$ was observed, even though fundamentally they measure two different things. The relationship is slightly weaker over the Catchment, where you see more memory in the $VOD_{MW}$ dataset as compared to the $LAI_{MD}$. This could be caused by a buildup of woody biomass, as this would theoretically be better detected with the $VOD_{MW}$ than with the $LAI_{MD}$. The period of sustained high $VOD_{MW}$ in the catchment during the 2008 to 2012 period aligns well with the $PR_{E5}$, which recorded 5 years of above-average rainfall over the Catchment. The ORD shows this increase above the climatology starting only the year afterwards (from 2009 to 2012), showing the lagged response of the system after a prolonged dryer period.

The $VOD_{MW}$ signal in the Delta is more complex:  the peaks in $VOD_{MW}$ do not coincide with prolonged time spans of high water availability, but seem to peak during shorter periods of increased water availability during overall conditions with medium to low $ODIA_{MD}$. This can be explained by what the VOD represents: in this case, it is related to biomass that is above the surface. During prolonged periods of high water, a larger extent of these regions are flooded. Therefore, within the 0.25° pixel, data that is not corrected for dynamic water bodies, the vegetation covered by these flooded areas might not be properly measured by the $VOD_{MW}$ signal. As it is also known that $VOD_{MW}$ values can be underestimated during flooded conditions (Bousquet et al., 2021). Note that the negative $SM_{MW}$ and ORD anomalies in 2019 have not led to the same intensity of vegetation decline, while in 2019 the $ODIA_{MD}$ was at a record low in the last 20 years. With the very strong relationship over the Delta between the anomalies of both $VOD_{MW}$ and $LAI_{MD}$ - two independent satellite-observed datasets - these observations very likely reflect the conditions on the ground. These results show that future use of even longer $VOD_{MW}$ records can help monitor complex regions like the Okavango Catchment and Delta. For example, following the progress on VODCA -which aims to build a data record similar to the ESA CCI SSM for $VOD_{MW}$ -future releases will also include the latest calibrated datasets as used here.

Three different sources of LST were tested over the Okavango Catchment and Delta. The highest correlation can be found between the $LST_{E5}$ and $LST_{MW}$, which most likely best represent the actual ground conditions. Although $LST_{MD}$ performs less well, the better correlation of $LST_{MD}$ against $LST_{E5}$ than $LST_{MW}$ might indicate that the overall best



performing dataset is the $LST_{E5}$. However, in many cases an observation-based long term dataset (e.g. the $LST_{MW}$) is still preferred. For example, in 2010 and 2011 the $LST_{MW}$ has the lowest temperature anomalies on record in the Delta, going to -3 K, while the $LST_{E5}$ remains more neutral. This is most likely caused by the lack of lateral water influx modelling from the ORD and following $ODIA_{MD}$ in ERA5-Land (Muñoz-Sabater et al., 2021), as shown in Section 4.4. The lack of moisture input into the model can lead to an underestimation of the latent heat flux and overestimation of the sensible heat flux, leading to an unrealistically high $LST_{E5}$.

In the Delta, 2015, 2016, and 2019 have been warm compared to the years before. The $LST_{MW}$ and $LST_{E5}$ both show that these are not unique occurences, as similar high values have been detected on multiple occasions before 2006. These seem to occur during periods of lower $ODIA_{MD}$, which shows dry anomalies of varying strength in these years. The catchment does see its highest and more prolonged peaks only in the last years, i.e. 2015 and 2019. These high peaks coincide with the strongest negative anomalies found for both $SSM_{MW}$ and $SSM_{E5}$, linking the high temperature and reduced moisture availability.

The precipitation-driven SSM in the Catchment aligns closely with $SSM_{MW}$ and $SSM_{E5}$ datasets. Especially in the period after 2010, the signal in the anomalies is very similar. Before 2010, it appears that the $SSM_{MW}$ shows slightly stronger dynamics than $SSM_{E5}$. In the Delta a mismatch is clearly seen between $SSM_{E5}$ and $SSM_{MW}$, especially with regard to the duration of the dry and wet peaks, but also in their intensity. With the knowledge that about 50%-75% of the water flux into the Delta comes from the ORD, and about 25%-50% from the PR, an analysis using Z-score anomalies was conducted to determine the driving signals behind the SSM anomalies, using the ORD, $ODIA_{MD}$ and PR as inputs. For $SSM_{E5}$, an almost one-to-one relationship was found with the PR, with little to no effects from the ORD or $ODIA_{MD}$. The $SSM_{MW}$ anomalies on the other hand, are almost equally driven by PR and $ODIA_{MD}$, which is much closer to the actual balance between the ORD and PR water fluxes for the Delta as expected from literature.

The almost one-to-one relationship between the $SSM_{E5}$ and $PR_{E5}$, and lack of signal related to the ORD due to the missing lateral water influx modelling, or alternatively dynamic open water bodies using the $ODIA_{MD}$, in ERA5-Land indicates that in a complex region like the Okavango Delta important forcings are missing. This for example could also cause the difference in $LST_{MW}$ and $LST_{E5}$ in 2010 and 2011, as the model cannot correctly convert the



incoming radiation into sensible and latent heat fluxes when the moisture conditions are inaccurate. On the other hand, while the $SSM_{MW}$ signal provides users with a better representation of total moisture conditions within the catchment, it can also not be interpreted as a pure SSM signal here, as it includes moisture information driven by the $ODIA_{MD}$. In a dynamic environment as the Okavango Delta, users should therefore clearly define what they require of such datasets to avoid unwanted side effects.


## 6    Conclusion

The anomalies of three different parameters, i.e. $SSM_{MW}$, $LST_{MW}$ and $VOD_{MW}$, were evaluated against other satellite-observed data sets and data from the ERA5-Land climate reanalysis. Although $SSM_{MW}$ and $SSM_{E5}$ correlate moderately well, structural differences were detected

over the Okavango Delta, where $SSM_{MW}$ contains a clear multi-year signal that is not in the $SSM_{E5}$. To determine the cause of this mismatch, an analysis was conducted to determine the impact of three sources of water into the Okavango Delta, i.e. the ORD, $ODIA_{MD}$ and the PR, on the SSM signal. The $SSM_{MW}$ signal appears to be driven about equally by the $ODIA_{MD}$ and the PR, while $SSM_{E5}$ is almost fully driven by the $PR_{E5}$. This indicates that ERA5-Land does

not properly include the lateral influx of the Okavango River, and therefore the use of $SSM_{MW}$ is preferred in this region.

For the $VOD_{MW}$, a direct comparison against $LAI_{MD}$ was made. Although the two parameters measure two different characteristics of the vegetation, good correlations. Over the Catchment, a stronger multi-year signal was detected in the $VOD_{MW}$, which could be related

to the build up of biomass, to which $VOD_{Mw}$ is theoretically more sensitive. For the Delta, both datasets are impacted by the increase in open water during long wet periods that can suppress the observed vegetation. This strong similarity as observed between the two datasets, indicate that it is very likely they are both representative for the in situ conditions.

$LST_{MW}$ was shown to be of good quality and correlated well with $LST_{E5}$ (>0.8). $LST_{MD}$ still managed to reach a significant correlation with $LST_{E5}$, but not with $LST_{MW}$, indicating that in general $LST_{E5}$ could be of highest quality of the three when looking at the temporal signal. However, at the record-low values in $LST_{MW}$ over the Delta in 2010-11, corresponding to the peak years of the ORD and $ODIA_{MD}$, it seems that $LST_{E5}$ cannot properly model the sensible





and latent heat fluxes because it is missing the lateral water component. This can have a large impact for detecting extremes, which are especially important in the current changing climate.

The findings of this research show the importance of not only relying on climate reanalysis,
but also the need for further development and maintenance of observational datasets like the ones derived from passive microwave observations. For example within the ESA CCI Soil Moisture datasets, but also the development of new CDRs on $VOD_{MW}$ like VODCA. Their ability to properly detect anomalies and extremes is very valuable in climate research, and can especially help to improve our insight in complex regions where the current climate
reanalysis datasets reach their limitations. With microwave data being available from 1978 onwards, the data can be used for long-term climate studies, near-real-time applications, e.g. monitoring complex natural systems like the Okavango Delta, and to constrain climate reanalysis through data assimilation techniques to overcome known model weaknesses.

# 7  Author Contributions

RS is the main author of this manuscript, and led the conceptualization, data curation, formal analysis, validation, visualization, and writing of the original draft. MV provided support for the conceptualization of the study and contributed to the writing and the visualization in Sect.
2, on the research area. CA supported this study by reviewing the manuscript and reforming the conceptualization to better put the research in perspective of the scientific community. WD contributed by extensively reviewing the manuscript. PW provided support in the understanding of the Okavango region, data provision for $ODIA_{MD}$, and reviewing of the manuscript. RJ was active in the conceptualization of the study and writing of the
Introduction.

# 8  Competing interests

The authors declare that they have no conflict of interest.



## 9  Acknowledgments

This study and the authors were supported by ESA's Climate Change Initiative for Soil
Moisture (Contract No. 4000104814/11/I-NB and 4000112226/14/I-NB).

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
