# Peer review of "Characterising natural variability in complex hydrological systems using Passive Microwave based Climate Data Records: a case study for the Okavango Delta"

_Hydrology and Earth System Sciences, 2021_

## Referee Comment (RC2)

**General comment**

This study analyses the natural variability of the water and soil-related component over the Okavango Delta (OD) region. The authors have used passive microwave (PMW) based Climate Data Record (CDR) obtained through inter-calibration of multiple PMW observations and a particular retrieval model. They compare the CDR to model outputs (ERA-5 Land), infrared-based estimates, and auxiliary datasets. The objective is threefold: 1) analyzing the complex hydrological region of the OD, 2) validating to some extent their CDR using ERA5, and 3) demonstrating the benefit of using PMW information to characterize processes that are not well implemented in the ERA5-L model such as the lateral flow-related soil moisture. The manuscript is well organized and well written, and I have appreciated its perspectives. Nevertheless, I believe additional work is needed to put this work in good shape for publication. In particular, some figures are needed to support statements of the authors and a few experiments could improve the analysis. I listed my comments below, which, I hope, will contribute to the preparation of an improved version of the manuscript.

**Majors comments:**

*1) Concerning the PMW CDR:*

- Too little information is given on the inter-calibration and retrieval model. I acknowledge that these two compounds are supported by published materials but the author could add information in an annex on the inter-calibration (such as the cost function used in the optimization during inter-calibration). Also a figure with raw (original) CDR time series over the two regions of interest (ROIs) before retrieving the climatological mean.

- In Table 1, the resolution of the pixel measured by PMW at the surface is needed as it is said that Brightness Temperatures (BT) have been aggregated at 0.25° to obtain gridded products.

- If descending orbit only have been processed such that it can be compared with MODIS, such information is needed p7-L192

*2) Concerning the LST analysis:*

- Land surface temperature from ERA5 (LSTe5) has been extracted only for the first layer (0-7cm), is there any information on the penetration depth from the PMW observation? The infrared MODIS-based LSTmd is used for comparison, as infrared LST has no penetration depth, how this could impact the analysis. Please comment on p12-L315.

- No Time series is plotted for LSTe5 and LSTmd. For a systematic analysis, these two must be added in Figure 4. It should better support the author's statement on LSTe5 through the manuscript (p22-L533; p23-L564) and in the abstract. This is not shown in the analysis yet.

- I would suggest adding LSTmd climatology in Figure 6

*3) Concerning the VOD*

- Climatology for ROi1 could be added in an annex to see if the LAI and VOD seasons are less correlated over catchment as it is stated p23-L595.

-Xband is less sensitive to leaves over dense forest, any experiment has been conducted in using/not using Xband for VOD in the omega-tau model?

*4) Concerning the Figures:*

- All scatterplots must have Xlabel and Ylabel for clearer reading. I suggest introducing correlation numbers inside the figures.

- All correlation numbers must have at most 3 digits as the 4th is not meaningful.

- "Absolute anomalies" in the title is misleading as "absolute" has another mathematical meaning. Replace by "raw"?

-add PRim in Figure 5 as well as PRe5 and PRim for RO1 (can be in annexes) as it is stated that PRe5 has high positive anomalies over the catchment (p21-L478) with no supporting information.

*5) Concerning the Linear regression experiment:*

- RMSE for Z-score is difficult to analyze, pleased replace by bias and std metrics in table2

- Please consider doing the Linear regression experiment for the catchment ROI1 to see if the SSM is more related to the precipitation upstream (as stated p22-L517).

- In the Table specify the considered ROI.

- It said that OIAD show some lagging from ORD, could you find optimal lag with cross-correlation between ORD-SSM and SSM-OIAD. This might lead to finding some buffering effect in SSM between ORD and OIAD.

**Minors comments :**

-p2-L36, miss-record
-p2-L51, "the The"
-p3-L63, BAMS, acronym is not defined
-p3-L93, PMW is not defined,
-p4-L111, use Section instead of "Chapter"
-p4-L114, LPRM is not defined yet E5L should be E5
-p7-L195, The sentence is misleading since not only the Xband is used
-p9-L224, what is an E-type gauge?
-p11-L289, VODCA is not defined
-add Z-score equation
-Caption of figure 3 seems misplaced (not attached to the figure on the same page)
-p21-L474, what "memory" replaces to buffer effect?
-p21-L489, SSM not SM

-In Fig6: +-15d is used for visualization only or to compute anomalies also? If it has been used for computing anomalies, this could lead to over-smooth the anomalies with the 90 days moving average window.

-p23-L534: could you be more specific. The increase of SMM with available solar energy, increases ET and avoids a false increase of LST but how is LSTe5 between 2011-2014?

 -p23-L553 verb is missing

---

## Author Response (AR1)

Dear editor,

In the revised version of the manuscript we have focused on improving the manuscript using the constructive suggestions as provided by the two referees. Below we summarized the content of the improvements and a few notes on other decisions we made concerning suggested changes.

- In general the text has been improved for readability. The referees highlighted several areas in the manuscript that needed some extra clarifications, which we followed up on in order to improve it. For example, this was done by adding some extra information on the intercalibration minimization function in Section 3.2.1 and better describing why we choose these datasets. The specifics can be found in the document with tracked changes.

- On the figures, Fig. 2 (EF), Fig. 3 (EF), and Fig. (4 CDEFGH) have been improved following the advice of the referees. The X/Y labels have been added and the correlations have been placed in the figures. Also, both in the figures and throughout the rest of the manuscript, correlations are now reported to two decimal places.

  After a discussion, we have decided not to follow up on the request to add the figure on the LST time series for ERA5, as we think Fig. 4F and the text sufficiently support the statement made and we want to keep the images to a reasonable limit that matches well with the text.

  A similar decision was made for the intercalibrated brightness temperatures. We appreciate the interest of the referee on this topic, however the focus of this study is on the resulting datasets, the climate data records, not on the brightness temperatures that go into it. If there were any issues with this data, this would be directly visible in the passive microwave derived anomaly time series as breaks. Therefore, we think the reference to Van der Schalie et al. (2021), which includes examples of the requested imagery, is sufficient. Of course, if the referee is interested in this work, he can always reach out and I am happy to share some results on X-band specific.

- The affiliation of three authors, including myself, have changed. As the company VanderSat was acquired by Planet at the end of 2021, the affiliation of R. van der Schalie, M. van der Vliet and R. de Jeu changed from VanderSat to Planet too.

We hope that the editor and referees are content with the improved manuscript provided, on both the applied changes and other choices made.

Kind regards,

Robin van der Schalie, on behalf of all authors

---

## Referee Report (RR1)

**General comment**

This study analyses the natural variability of the water and soil-related component over the Okavango Delta (OD) region. The authors compare their surface soil moisture, land surface temperature, and vegetation optical depth Climate Data Record (CDR), previously derived using passive microwave (PMW), to model outputs (ERA-5 Land), infrared-based estimates, and auxiliary datasets. The objective is threefold: 1) analyzing the complex hydrological region of the OD, 2) validating to some extent their CDR using ERA5, and 3) demonstrating the benefit of using PMW information to characterize processes that are not well implemented in the ERA5-L model such as the lateral flow-induced soil moisture.

The authors have provided valuable responses to all of my previous comments and questions and the readability of the text, numbers, and figures has been improved in the revised version of the manuscript. **If I am looking forward to reading their answer to the following comments, I consider the manuscript ready for publication after technical revision.**

I acknowledge here their negative response regarding my suggestions to add:
1) a figure with the original (non-anomaly) data in the appendix.
2) the figure on the LST time series for ERA5
If I think such information is useful for a broader view of the analysis, I will not insist on these points as this might imply enlarging the article in losing its key message on anomalies.

**Two comments on the author's response :**

1) Some of my comments had received positive answers from the author's reply but were not implemented in the revised manuscript with no additional information.
see the following point :

*-Add PRim in Figure 5 as well as PRe5 and PRim for RO1 (can be in annexes) as it is stated that PRe5 has high positive anomalies over the catchment (p21-L478) with no supporting information.*
 -To make sure all the information is included in the figures, we will include the PRE5 (ROI1) and PRIM (ROI1/2)in Figure 5 so it includes information on all support data.
Information on the final author's decision concerning this point is needed.

2) I would like to clarify a point regarding the inter-calibrated brightness temperature. In their answer, the authors state that they decide to not include a figure depicting inter-calibrated brightness temperatures. To my knowledge, no such comment was raised in the first round of review. This confusion might be due to the lack of clarity in my first comment from the first round in which I deal with both information on the inter-calibration and the original CDR (i.e. retrieved variable) at once.

**Comments on the revised manuscript:**

- Equations 1, 2,3 are empty, white boxes replace variable names.

- l.19 : 'long-term'
- l.60 : 'are' described
- l.78 make's'
- l.98 'The' republic of Botswana
- l.114 'the' evaluation ….
'and of' sounds awkward
- l. 125 'an' actively inundated…
- l.141 In this study','
- l.155 a growing 'water demand'
- l.156 and 'was' likely
- l.183 'nighttime'
- l.188 unfortunetly','
- l.213 analysis','
- l.218 ','the layer
- l.223 period 'from'
- l.231 'long-term'
- l.234 '1998-2020' period
- l.237 'derived from SWIR' or 'derived by using SWIR'
- l.241 'up-to-date'
- l.243 per unit 'of' ground
- l.259 for this study','
- l.269 delete : from the soil and 'the' from …
- l.313 'long-term'
- l.324 representative 'of' the moisture
- l.326 what cause 'them'
- l.329 'dataset'
- l.343 'nighttime'
- l.350 '91-day'
- l358 'the' following section
- l.366 'datasets'
- l.368 'a first step in this' sound awkward for the beginning of a paragraph
- l.378 delete as 'for'. Furthermore what is the subject of can ?
- l.393 : time 'serie' to avoid redundancy
- l.444: 'breakaway'
- l.478: 'long-term'
- l.546: 'occurrences'
- l.573: 'such' as the Okavango
- l.600: 'the 'highest quality

---

## Author Response (AR2)

Dear editor,

I would like to thank the reviewer for its supportive comments and the in depth review. We have corrected all the textual errors that were identified by the reviewer.

The latest round also helped clarify some of the comments of the first review round. In order to address his worries, both on the precipitation and land surface temperature information, we have added the requested figures to the Appendix (Sect. 11).

Figure A1 in the Appendix now contains the missing land surface temperature anomaly time series from ERA5 and MODIS. We inform the reader about this at the start of Section 4.3. F

Figure A2 contains the precipitation information from both ERA5 (Catchment) and IMERG (Catchment and Delta). The reader is directed to this extra information at the start of section 4.4.

So the issue on the lack of supportive information of statements made in the Discussion (Sect 5) is now resolved.

Please let us know if there are any other points that you would like to see addressed, so that we can publish the most complete version of the manuscript.

Kind regards,

Robin van der Schalie, on behalf of all authors